# Evolution of the Spatial-Temporal Pattern and Social Performance Evaluation of Community Sports and Fitness Venues in Shanghai

**DOI:** 10.3390/ijerph19010274

**Published:** 2021-12-27

**Authors:** Feng Sun, Jinhe Zhang, Jingxuan Ma, Chang Wang, Senlin Hu, Dong Xu

**Affiliations:** 1School of Geographic and Oceanographic Sciences, Nanjing University, Nanjing 210023, China; sf9300@163.com (F.S.); mg20270048@smail.nju.edu.cn (J.M.); wangchang@nju.edu.cn (C.W.); xudong_njnu@163.com (D.X.); 2Huangshan Park Ecosystem Observation and Research Station, Ministry of Education, Huangshan 245899, China; 3The Center for Modern Chinese City Studies, School of Urban and Regional Science, East China Normal University, Shanghai 200062, China; hsllh520@163.com

**Keywords:** community sports and fitness venues, temporal and spatial evolution, social performance

## Abstract

The study of the spatial-temporal pattern and social performance of urban public services is a basic task for achieving urban fairness and justice. Through spatial analysis and social performance evaluation, this study explores the evolution of spatial-temporal patterns and the social performance of community sports and fitness venues in Shanghai from 1982 to 2019. The results show that the construction of Shanghai’s community sports and fitness venues presents the evolution pattern of “urbanization-suburbanization-reurbanization”. The center of construction has always been in the urban area and first moved toward the south and then toward the north. Government investment was the main source of funds for the construction of venues, and social investment has been steadily growing. The number and area of multiple types of venues has increased significantly, including trails, gymnasiums, and courts. The overall service coverage radius of Shanghai’s community sports and fitness venues has been significantly increased, and the regional equality between the core and peripheral areas has been obviously improved. The overall per capita service location entropy has not been significantly improved. The old city center and the peripheral area have always been the low-value areas, and the old city center is surrounded by high-value areas. The “low-high-low” three-circle spatial structure continues to exist, but around the old city center, the scope of the high-value area has expanded markedly. There was a significant optimization of social performance from 1999 to 2009. The social performance of the community sports and fitness venues in urban areas is better than that in suburban areas, but the optimization of social performance in suburban areas is greater than that in urban areas. The above analysis is expected to provide references for rationally arranging urban sports and fitness spaces, enhancing the fairness of urban public services, improving the quality of residents’ lives, and assisting the implementation of the “Healthy China” national strategy.

## 1. Introduction

Modern sports are modernistic products of the industrial revolution, and industrialization and modernization mainly occur in urban space; thus, there is a natural internal connection between urban and modern sports [1]. With the rapid development of urbanization, individuals are facing high-intensity work pressure and the alienation of interpersonal relationships in contemporary society. Thus, individuals are eager to find self-identity through some channels, and in this context physical sport becomes an important way for modern people to achieve individual liberation and self-expression [2,3]. Sports are increasingly welcomed by urban residents, and sports space has become a significant part of urban planning. By promoting the development of “urban sportization”, the relationship between urban areas and sports is becoming closer [4]. Community sports, one form of urban sports, have become an indispensable and important part of urban construction, due to their popularity, benefits to public welfare, and proximity to homes. Community sports can provide urban residents opportunities for physical fitness, entertainment and leisure, and social interaction. The development level of community sports has become an important indicator of urban qualities [5,6]. In 2016, the “Healthy China 2030 Planning Outline” was issued by the Central Committee of the Communist Party of China and the State Council, which proposed that the public service system for nationwide fitness should be improved and that nationwide fitness activities should be widely generalized. In 2019, the State Council published the “Healthy China Action (2019–2030)”, which further called for promoting the construction of a basic public sports service system, coordinating the construction of national fitness facilities, and creating a “15 min fitness circle” for the public. Since community sports and fitness venues are important urban public service facilities, increasing their number, optimizing their spatial layout, and realizing social fairness and justice in sports fitness services have attracted much attention.

Currently, academic research on the spatial layout of sports and fitness venues mainly includes four categories. First, the service scope [7], spatial pattern [8], spatial evolution [9], and influencing factors [10] of different types of sports and fitness venues were explored at different scales by using spatial analysis methods. Some studies have revealed that population characteristics, government policies, the characteristics of the leisure industry, transportation accessibility, and participants’ interests can affect the planning of urban sports facilities [11,12,13,14,15]. Second, there are differences and causes of the spatial distribution of sports and fitness venues among different regions, such as between countries [16] and between urban and rural areas [17]. It was found that there is a strong correlation between the frequency of sports activities and the number of swimming pools in metropolitan municipalities, while the number of athletic fields is more important in medium-sized municipalities in Germany [18]. Third is the spatial justice of sports and fitness venues and the fitness needs of disadvantaged groups, such as ethnic minorities [19,20], women [21,22], the LGBTQ+ community [23], immigrants [24], and the disabled [25]. Data are collected using questionnaires, interviews and other methods to obtain their choice preferences, travel radius, and frequency of participation to provide information for the rational layout of sports and fitness venues. Research has shown that adolescent participation in sports has a positive correlation with the surrounding social capital in the Netherlands [26]. Other studies have shown that increasing the proportion of nonprofit leisure sports facilities around the community has a significant effect on improving the frequency of population participation in sports activities and can also effectively solve the inequality of opportunities in the process of sports participation [27]. Fourth, building sports and fitness venues has a positive effect on urban space renaissance [28], urban image shaping [29], and local social construction [30].

Additionally, the research perspective on the layout of public service facilities has also evolved. In the 1960s, Teitz’s location theory of public service facilities laid a theoretical foundation for the research on the layout of public service facilities and promoted the systematic development of relevant research. Based on Teitz’s basic system theory paradigm, research on the layout of public service facilities initially started with econometric and behavioral geography, with a focus on facility efficiency [31,32,33]. Markets gradually alternate the state’s role in public services because of the government’s difficulties in optimizing supply alone, i.e., the national model has shifted to the marketization model to supply public services. Meanwhile, the appearance of some social problems like spatial injustice and social differentiation in western countries results in the turn of scholars’ research trends. For example, Harvey, the pioneer of Marxist geography who emphasized the significance of political geography and national social theory in the spatial service of public amenities, used to be a representative of positivism [34,35,36]. Since the end of the 20th century, western scholars have reflected on the previous welfare system and promoted civil rights and democratic values to a higher level in the field of public services. The results of exploring the use preferences of residents, from the perspective of sociology, to provide support for the layout of public service facilities emerged a lot, which developed the humanistic research on the layout of public service facilities [37,38]. Moreover, researchers further analyzed the rationality of the layout from the perspective of welfare economics based on the traditional location theory, realizing the optimization of the layout of public service facilities [39,40].

The evaluation of the social performance of the distribution of urban service facilities is crucial for achieving social equity and justice in urban services. Social equity and justice involve complex concepts such as equality, equity, and justice [41]. Equality is a fundamental value orientation and the highest moral code, as well as an important value dimension pursued by modern society and political systems [42]. Equalization is the basic issue of equality theory; that is, in which aspects should people be equal to each other [43]? Rawls and Dworkin support the idea of equality of resources, and they care about the equality of external material resources, such as individual income and wealth [44]. Rawls put forward the principle of fair equality of opportunity, proposing that society should provide equal working opportunities and equal social status for those with the same ability, skills, and motivations. Rawls’s principle emphasizes the concept of equality for all [45]. This concept coincides with the “welfare state” system commonly practiced in western developed countries from the Second World War to the 1970s. During this period, urban planning emphasized equal urban services for all, and spatial equality was the core topic at this stage. Per capita ownership of service facilities on a large scale was often used as the evaluation standard of social performance [46]. Equity refers to the reasonable distribution of rights and benefits. It is a reasonable design and ideal arrangement for the relationships between members of society and calls for fair distribution of wealth and income through fair rules and systems [47,48]. In the 1970s–1990s, to improve management efficiency under the social welfare system, developed western countries adopted the “new public management” reform [49]. During this period, urban services research turned to spatial equity. By using spatial analysis tools, such as GIS, attention was given to the location and accessibility of facilities and their service benefits to the public [50,51]. Technology development and different understandings enhanced the institutionalization of the evaluation of social performance in the distribution of urban service facilities at this stage, representing more social equity. Justice is a broad moral and political ideal. Rawls put forward the theory of “justice as fairness”, emphasizing that we should pay attention to the rights and entitlements recognized and performed by individuals, while fairly distributing the benefits and burdens in society. The distribution of benefits should be more inclined to the most vulnerable or disadvantaged groups to maximize the benefits of the beneficiaries most in need [52]. Since the end of the 20th century, the emergence of the concept of “new public service” in urban planning has implemented the idea of “justice as fairness” [53]. Social justice as a concept has been widely considered by advocating respect for diverse needs and paying attention to the benefits of vulnerable groups [54,55]. In general, the practice of social performance evaluation in the distribution of urban service facilities roughly experienced three stages: from emphasizing “regional equality” to “social equity” to “social justice” [56,57,58]. Of these, regional equality is the concept that focuses on the difference in services, specifically, the ownership of different service facilities among different regions. Social equity emphasizes the service efficiency of facilities, represented by the spatial matching between service quantity and quality and population distribution. Social justice calls for the distribution of facilities for vulnerable groups.

In general, most of the current research focuses on the spatial pattern of regional sports and fitness venues in a certain historical period. There are few studies on their temporal and spatial evolution over a long-term period and even fewer on the evaluation of their social performance from the urban public service aspect.

This paper used 230 subdistricts and towns in Shanghai as research units to analyze the temporal and spatial evolution of community sports and fitness venues from 1982 to 2019. Data such as population and housing in 1999, 2009, and 2019 were collected. This study measured the social performance of Shanghai’s community sports and fitness venues based on regional equality and social equity aspects.

This paper tries to answer the following questions:(1)What were the temporal and spatial evolution patterns of community sports and fitness venues in Shanghai?(2)What were the differences in the social performance of community sports and fitness venues during different historical periods?(3)What were the impacts of urbanization on the social performance differences in community sports and fitness venues between urban and suburban areas?

Through the above analysis, it is expected that this study will provide useful information for developing countries, with political and cultural backgrounds different from the west, to rationally arrange sports and fitness venues. Moreover, the social performance evaluation system of urban public facilities will be established, which can contribute to making a comparative study of different periods or different cities and overcoming the urban injustice of public service.

## 2. Research Design

### 2.1. Research Subject

Sports and fitness venues are the recreation places that provide the public space for fitness and facilities services. The community sports and fitness venues in this paper are the places that meet the residents’ daily basic sports and fitness needs with a relatively small service scope, including citizens’ educational fitness centers, trails, gyms, courts, etc. The ‘Healthy China Action (2019–2030)’ proposed creating a ‘15 min fitness circle’ for the public. A 15 min walk was used as the service radius of community sports and fitness venues. Previous studies showed that a 15 min walking distance can be converted into a 700 m straight-line distance [59]. This paper used 700 m as the buffer zone to show the spatial relationship between the community sports and fitness venue and the residential district directly in 1999, 2009, and 2019 (Figure 1). The more buffer zones the residential district was covered by, the more opportunities the residents had for sports and fitness.

### 2.2. Research Area

Shanghai is one of the most economically developed modern international metropolises in China, which promotes the integration of the Yangtze River Delta and the development of the Yangtze River Economic Belt. As the region with the highest level of urban management in China, Shanghai has been promoting the construction of urban sports and fitness venues for a long time. In the 1990s, taking the opportunity to host major sport events, Shanghai successively built a number of multifunctional stadiums, such as Shanghai Stadium, Shanghai International Circuit, Shanghai Hongkou Football Stadium, and Oriental Sport Center. In addition, Shanghai successively issued the ‘Shanghai Municipal Citizens’ Sport and Fitness Regulations’ (2000), the ‘Shanghai Youth Football Elite Training Base Construction Plan’ (2015), the ‘Shanghai National Fitness Implementation Plan (2016–2020)’ (2016), ‘Guidance of Hosting National Fitness Events in Shanghai’ (2017), ‘Suggestions on Accelerating the Innovative Development of Sports Industry in Shanghai’ (2018), ‘Shanghai’s 2019 Health China Tour Event Plan’ (2019), and a series of policies and measures related to physical fitness. In 2018 alone, Shanghai built 89 trials, newly built and rebuilt 72 stadiums, and built 342 citizen educational fitness centers. The construction of Shanghai’s sports and fitness venues ranks among the top cities nationwide. It is of great value to study the temporal and spatial evolution of Shanghai’s community sports and fitness venues. One the one hand, it is helpful for the dynamic analysis of the relationship between sports and urban spaces by studying the temporal and spatial evolution. On the other hand, through tracing the development path of community sports and fitness venues in Shanghai, this paper will provide references for cities with high population densities in other developing countries to achieve sustainable urban planning and urban governance.

### 2.3. Data Source and Processing

The community sports and fitness venue data were collected from Shanghai’s community sport facility management service platform (http://www.shggty.com.cn/facilityMap.html, accessed on 26 October 2020). After data filtering, cleaning, and space matching, the community sports and fitness venue data were obtained: 483 in 1999, 7475 in 2009, and 16,906 in 2019.

The urban spatial administrative boundary data of Shanghai came from the ‘Shanghai Atlas’ published by Planet Map Publishing House in 2018. ArcGIS 10.2 software (developed by Environmental Systems Research Institute, Inc., Redlands, CA, USA) was used to convert the base map data.

The data on the boundary of Shanghai’s districts came from the Humanities and Social Sciences Big Data platform, founded by East China Normal University (http://sdsp.ecnu.edu.cn/data/dv/shxqbjsj;jsessionid=def1de85ccc636c190759815e241, accessed on 20 October 2020). The data were collected in 2018. After spatial matching, the residential districts were obtained: 3345 in 1999, 6795 in 2009, and 7428 in 2018. A residential district refers to a region with a certain population size, surrounded by urban trunk roads or natural boundaries, that has relatively perfect public service facilities to meet the basic material and cultural needs of residents. The residential district is the lower administrative unit of the street or town.

The permanent populations in 1999 and 2009 were derived from the fifth (2000) and sixth (2010) Chinese National Population Census. The permanent population in 2017 was derived from the 2018 Statistical Yearbook and Development Statistical Bulletin of Shanghai’s districts and towns. Subdistricts and towns were used as basic demographic units (Figure 2).

Due to availability, a data mismatch in terms of collection time exists, which may affect the accuracy of the research. Specifically, community sports and fitness venue data were from 2019, residential area data were from 2018, and population data were from 2017.

However, generally speaking, indicators such as permanent populations and residential communities are usually stable over a short period of time. In addition, this study used small and medium-sized subdistricts and towns as the research units, which can reduce the unreliability of the data.

### 2.4. Research Method

#### 2.4.1. Spatial Analysis Methods

Kernel density estimation and standard deviational ellipse analysis were used to explore the spatial pattern and evolution law of Shanghai’s community sports and fitness venues at different urbanization stages.

(1) Kernel density analysis. To visually show the cold and hot areas of community sports and fitness venues, the kernel density analysis method was used. The points in the area were assigned different density weights to make the distribution of density attribute results in the area smoother and more intuitive. The mathematical formula is [60]:(1)$$f\left(x\right)=\frac{1}{nh}\overset{n}{\underset{i=1}{\sum }}K\left(\frac{{S-S}_{i}}{h}\right)$$

In this formula, *K*() is the kernel density equation, *h* is the bandwidth (*h* > 0), *n* is the number of samples, and *S−S_i_* is the distance from the spatial sample point, *S_i_*, to the estimated point, *S*.

(2) Standard deviational ellipse analysis. The standard deviational ellipse analysis module in ArcGIS 10.2 was used to study the central trend, discrete trend, and direction trend of the distribution of community sports and fitness venues in Shanghai. The ellipse with a standard deviation of 1.0 contained 60% of the sample. The long and short axis directions are related to the standard deviational ellipse shape. The long axis and the short axis represent the maximum and minimum diffusion directions, respectively. The dispersion degree of the distribution of community sports and fitness venues was determined by the area of the standard deviational ellipse. The smaller the area was, the closer the distribution was to the center of gravity [61].

#### 2.4.2. Social Performance Evaluation Method

Following the social performance evaluation methods of urban public services, such as green space and transportation [62,63], the social performance evaluation system of community sports and fitness venues was constructed from the dimensions of ‘regional equality’ and ‘social equity’, based on accessibility research. The dimensional distribution was measured by the indicators of ‘service radius coverage’ and ‘per capita service location entropy’. Regional equality focuses on the differences in service levels in different regions, and social equity emphasizes the spatial matching of service levels and the population distribution. Current urban public service accessibility is often measured by the buffer analysis method [64,65]; thus, this paper used buffer analysis to measure the accessibility between community fitness venues and residential communities.

(1) Service radius coverage rate. The service radius coverage rate was used to measure the spatial equality of regional community sports and fitness venues. The formula is:(2)$$R_{i}=\frac{\sum_{j=1}^{n}s_{j}{-S}_{a}}{S_{i}}$$

In this formula, *R_i_* is the coverage rate of the service radius of community sports and fitness venues in unit *i*, *s_j_* is the area of the community covered by the service radius of community sports and fitness venue *j*, *n* is the number of community sports and fitness venues that can service the community in the unit, *S_a_* is the overlap area of the community covered by the service radius of the community sports and fitness venues, and *S_i_* is the area of the community in unit *i*.

(2) Per capita service location entropy. Drawing on the location entropy method [58,66,67], the per capita service location entropy was used to measure the spatial matching between the number of community sports and fitness venues and the permanent population. The formula is:(3)$${LD}_{i}=\frac{Z_{in}{/P}_{i}}{Z_{an}{/P}_{a}}$$

In this formula, *LD_i_* is the per capita service location entropy of community sports and fitness venues in unit *i*. An *LD_i_* less than 1 means that the per capita possession of community sports and fitness venues in unit *i* is less than the overall level. Otherwise, it is higher than the overall level. *Z_in_* is the area of the community covered by the effective service radius of community sports and fitness venues in unit *i*, *P_i_* is the number of permanent populations in unit *i*, *Z_an_* is the area of the community covered by the effective service radius of community sports and fitness venues in the entire area, and *P_a_* is the number of permanent populations in the entire area.

## 3. Results

### 3.1. Evolution of the Spatial and Temporal Patterns of Community Sports and Fitness Venues in Shanghai

Because there were few community sports and fitness venues in the 1980s [68], this paper divides the time period into three phases: 1982–1999, 2000–2009, and 2010–2019. To explore the evolution of the overall spatial pattern of Shanghai’s community sports and fitness venues more intuitively, Shanghai is divided into two areas, urban areas and suburban areas, using outer ring elevated roads as the boundary (Figure 3).

Figure 3 shows that the evolution of the construction of community sports and fitness venues in Shanghai follows a pattern of “urbanization-suburbanization-reurbanization”. Specifically, from 1982 to 1999, Shanghai built 483 community sports and fitness venues, mainly in the Jiangning Road subdistrict (16.32/km^2^), Hudong New Village subdistrict (6.8/km^2^), and Gonghe New Road subdistrict (3.5/km^2^) within the urban area. This shows that the high-density regions have small spatial areas; spaces are fragmented, there are far fewer community sports and fitness venues in suburban areas, and the distribution between urban areas and suburban areas is significantly different. From 2000 to 2009, there were 6992 community sports and fitness venues built in Shanghai, mainly in the urban area. However, the area of high-density regions became larger, and there was a tendency for the distribution to spread to suburban areas. High-density areas appeared in the southwest of urban areas, including Hongqiao Town, Xinzhuang Town, and Qibao Town. Although the urban area was still a hot spot for venues, the number of venues increased in suburban areas and the gap between urban areas and suburban areas narrowed. From 2010 to 2019, Shanghai built 9431 community sports and fitness venues. The construction of high-density areas was concentrated in the urban area, for example, the Laoximen subdistrict (23.07/km^2^), Caoyang Xincun subdistrict (21.43/km^2^), and Caojiadu subdistrict (19.78/km^2^). The gap between the urban areas and suburban areas at this stage also improved, but the high-density area contracted, representing a reurbanization phase.

To further analyze the evolution of Shanghai’s community sports and fitness venues, the standard deviational ellipse method was used to explore the shift of the gravity center of the overall spatial distribution (Figure 4). Figure 4 shows that from 1982 to 1999, the center of the construction of Shanghai’s community sports and fitness venues was in the Dapuqiao subdistrict in the urban area. From 2000 to 2009, the center moved to the southern suburban areas: the Shanggang New Village subdistrict. From 2010 to 2019, the center of construction moved north again to the Nanjing West Road subdistrict in the old city. In summary, during these three phases, the center of the construction of Shanghai’s community sports and fitness venues moved first toward the southern suburban areas and then toward the northern urban area.

### 3.2. Evolution of the Attributes of Community Sports and Fitness Venues in Shanghai

To further explore the evolution of community sports and fitness venues in Shanghai, this paper investigated the attributes of community sports and fitness venues at different stages, including funding sources, types, and areas (Figure 5).

First, we address the funding source. From 1982 to 1999, the main source of funding for community sports and fitness venues in Shanghai was the public welfare lottery fund. From 2000 to 2009 and from 2010 to 2019, the main funding sources were municipal finance. During the second and third stages, the number of community sports and fitness venues funded by subdistrict finance decreased. However, the number of community sports and fitness venues invested in and constructed by municipal finance, district finance, town finance, and village finance from 1982 to 2019 has been growing. Thus, government investment has always been the main source of funding for the construction of community sports and fitness venues in Shanghai. In addition, the number of socially invested venues from 2000 to 2009 increased by 20 times compared to 1982 to 1999. The number of social-capital-invested venues from 2010 to 2019 increased by 1.67 times compared to 2000 to 2009. Data indicate that the rapid development of the real estate industry promoted an increase in the amount of social capital invested in community sports and fitness venues.

Second, we discuss the type. From 1982 to 1999, the citizen’s educational fitness center was the most important type, accounting for more than 85% of the total. The reason for this may be related to its small area, low construction cost, and low space requirement. After 2000, the number of different types of community sports and fitness venues increased significantly, including trails, gyms, and courts, reflecting that diverse and personalized sports and fitness needs were considered. In terms of spatial distribution (Figure 6), the citizen’s educational fitness center was distributed more and more evenly in the three phases, showing the characteristics of spreading from urban areas to suburban areas. A large number of courts emerged in the suburban areas from 2000 to 2009, while there were many trails built evenly throughout the region from 2010 to 2019.

Third, we address the area. The average areas of community sports and fitness venues in 1982–1999, 2000–2009 and 2010–2019 were 141.46 m^2^, 198.53 m^2^ and 203.92 m^2^, respectively. The construction area continued to increase, which may better meet the residents’ sports and fitness needs.

### 3.3. Evolution of the Social Performance of Community Sports and Fitness Venues in Shanghai

This study used the service radius coverage rate and per capita service location entropy to characterize the regional equality and social equity of Shanghai’s community sports and fitness venues in 1999, 2009, and 2019 and explored the evolution of its social performance (Figure 7).

In terms of regional equality, the overall service radius coverage rate of community sports and fitness venues in Shanghai was significantly improved. The spatial equality between the core and peripheral areas was significantly improved. The change was more significant during the period from 1999 to 2009. Specifically, in 1999, 2009, and 2019, the average service radius coverage rates of community sports and fitness venues in Shanghai were 0.33, 0.84, and 0.89, respectively. The overall service radius coverage rate was growing. In 1999, the distribution of the service radius coverage rate was “high in the center, low in the peripheral area”. The old city centers, such as the Nanjing West Road subdistrict, the Jing’an Temple subdistrict, and the Caojiadu subdistrict, were high-value concentrated areas. In 2009, the pattern of “high in the center, low in the peripheral area” no longer existed. Except for the western part of the region and subdistricts and towns in Chongming District, the service radius coverage rate in most areas of the region, as a whole, reached 0.6 or more. Units with a coverage rate above 0.8 accounted for 80%. Regional equality was greatly improved. In 2019, the regional equality was further optimized. The service radius coverage rates of Xuxing Town, Sheshan Town, and Sijing Town in the west increased significantly, with streets and towns above 0.8 accounting for 88.7%. The development of urbanization has greatly improved the regional equality of Shanghai’s community sports and fitness venues.

In terms of social equity, the overall per capita service location entropy of Shanghai’s community sports and fitness venues has not improved significantly. The old city center and the periphery have always been low-value areas, and the old city center is surrounded by high-value areas. The ‘low-high-low’ three-circle spatial structure continues to exist, but the high-value area surrounding the old city center has expanded significantly, and the degree of optimization from 1999 to 2009 is greater. Specifically, the average per capita service location entropy of Shanghai’s community sports and fitness venues in 1999, 2009, and 2019 were 1.11, 1.08, and 0.91, respectively. The overall per capita service location entropy did not change much and showed a slight downward trend. The cause may be related to the continuous growth of Shanghai’s permanent population, indicating that the construction of community sports and fitness venues lags behind the growth of the permanent population. In 1999, the distribution of per capita service location entropy roughly showed a three-circle pattern of “low-high-low”. Densely populated central old subdistricts such as Nanjing East Road Street, Huaihai Middle Road Street, and Xujiahui Street were low-value areas, while Qibao Town, Beicai Town and Kangqiao Town were high-value areas in the surrounding areas of the old city center, and outer suburban areas were also low-value areas. In 2009, the “low-high-low” three-circle pattern was still maintained, but the proportion of units with per capita service location entropy greater than 1 rose from 27.83% 10 years ago to 34.35%, and the size of the high-value areas around the old city center increased significantly. In 2019, the three-circle pattern of per capita service location entropy did not change much, and the high-value area around the old city center further extended westward.

### 3.4. Comparison of the Social Performance of Community Sports and Fitness Venues between Urban and Suburban Areas

This study compared the social performance of urban and suburban community sports and fitness venues using the mean of the service radius coverage rate and per capita service location entropy of each unit during different time periods, as shown in Table 1.

First, the social performance of community sports and fitness venues in urban areas was better than that in suburban areas. In different historical periods, the service radius coverage rate and per capita service location entropy of community sports and fitness venues in Shanghai were both better in urban areas than in the overall area, while the indicators in the area as a whole were better than those in suburban areas, indicating that the regional equality and social equity of community sports and fitness venues in urban areas were better than those in suburban areas. Second, the optimization of social performance in suburban areas was greater than that in urban areas. From 1999 to 2009, the service radius coverage rate and per capita service location entropy of suburban areas increased by 323.53% and 5.81%, respectively. The service radius coverage rate of urban areas increased by 84.91%, while the per capita service location entropy decreased by 8.57%. Data show that social performance in suburban areas has improved more than that in urban areas.

## 4. Discussion

The research subject of this paper is the community sports infrastructure that is required to meet the daily needs of residents and nearby residential areas. The layout of sports infrastructure is not only related to the residents’ quality of life and the establishment of the national image, but it is also related to sustainable urban planning and urban governance. The research on the location selection and spatial effect of sports infrastructure is an issue of concern for scholars from different cultural backgrounds.

Since World War II, with the rise of civil society in western countries, the social demand and scope of supply of sports infrastructure has been expanding, and the supply mode of sports infrastructure has undergone great changes. In the 1960s, Cooper L and Teitz M successively put forward the location theory of urban public facilities and created a new field of geographical research [69,70]. In the 1970s, social differentiation, location efficiency, and spatial equity attracted extensive attention from western scholars [71]. As an important urban public service facility, sports infrastructure was paid more and more attention at this stage [72]. The research showed that under the comprehensive influence of multiple forces, such as politics, economy, and culture, the welfare state system in western countries has gradually faded out and the main body of public service supply has been diversified. The spatial distribution of the supply of urban public facilities is more favorable to high-income people, and the supply and demand show an “Inverse-care law” [73].

The large-scale construction of urban public facilities in China began in the 1980s. After experiencing unprecedented large-scale and high-speed “urbanization” in the world, China has become the largest investor in urban public facilities in the world. After more than 40 years of development, the transition of the Chinese infrastructure’s allocator was from a centralized administrative order to a market mechanism. The supply subject and supply mechanism of urban public facilities have gradually diversified, which resulted in the distribution of urban public facilities from spatial equilibrium to spatial differentiation. Under political and economic conditions different from the west, the evolution of the spatial pattern of urban public facilities and the resulting issue of social equity and justice in China, which is in the period of social transformation, deserve our discussion. However, limited by data and methods, few researchers are focusing on the layout of urban public facilities in developing countries (such as China) due to a long time scale, a lack of theoretical discussion and empirical analysis, and because there is less research to evaluate their social performance combined with socio-economic data.

The construction of sports and fitness venues in Shanghai also began in the 1980s. At that time, China had just experienced reform and was opening up. To enhance people’s national self-confidence and promote China’s international image, the Chinese government began to pay attention to the development of competitive sports, represented by the Olympic Games. Combined with its advantages, Shanghai focused on the development of Olympic events such as track and field, swimming, and rhythmic gymnastics. During this period, the administrative order was the main allocator of sports resources, which caused the development of competitive sports to be significantly faster than that of mass sports. Many large stadiums were built or planned, while the number of community sports and fitness places was small. In the 1990s China entered the stage of deepening reform, and the government established the reform goal of building the socialist market economic system in 1992. At that time, Shanghai’s sports system changed from a highly centralized sports system under the planned economic system to a modern sports system under the market economic system, and relevant sports laws and regulations began to be formulated. In 1995, Shanghai put forward “Shanghai’s National Fitness implementation plan” and established Shanghai’s leading group of national fitness, which greatly promoted the construction of community sports and fitness venues. In the 2000s Shanghai’s government put forward the goal of building Shanghai into a first-class sports center city in Asia. In 2000, Shanghai promulgated the “Regulations on sports and fitness for citizens in Shanghai”, the first local law on mass fitness in the Chinese mainland, which promoted the perfection of the sports law system. Shanghai took the lead in putting forward the concept of weaving sports and fitness into daily life, which further developed community sports and fitness venues during this period, with the construction scope spreading from urban areas to suburban areas. Taking the citizen’s educational fitness center as an example, the construction of Shanghai’s community sports and fitness venues achieved full coverage of streets and towns by 2009. In the 2010s China’s government put forward the concept of a “15 min fitness circle”, which showed that sports and fitness have been integrated into residents’ daily life. Moreover, in 2015, in order to improve the living environment in the old urban area, the government proposed the concept of “Double Urban Repairs”, including two programs of “Ecological Restoration” and “City Betterment”, which significantly improved the infrastructure in Shanghai’s old urban area. At this stage, the number of community sports and fitness venues continued to increase, and the layout of the venues showed the characteristic of “reurbanization”.

By reviewing the research on the layout of urban public facilities and the development process of Shanghai’s community sports and fitness venues, the main differences from previous studies were as follows: (1) Visualization of the spatial pattern of Shanghai’s community sports and fitness venues in different periods, which contributed to a dynamic analysis. (2) Construction of the social performance evaluation system to quantify the equity and justice of urban public facilities’ layout. (3) Visualization of the spatial-temporal evolution of sports infrastructure’s social performance in Shanghai and the comparison of the differences between urban areas and suburban areas. Through the above analyses, on the one hand, attention was paid to the urban construction in developing countries under a political structure and cultural order different from the west, which enriched the research on the layout of urban public facilities and contributed to the solution of urban spatial injustice. On the other hand, this paper attempted to establish the social performance evaluation system of urban public facilities to that has systematic and universal applications, which helped conduct a diachronic comparison (comparison of different periods of the same city) and a synchronic comparison (comparison of the same period of different cities).

Of course, there are some limitations to this paper. This research covered a period over nearly 40 years and the research unit is the small scale of streets and towns. Due to the lack of standardization of official data statistics in the early years, the socio-economic data were missed severely. Therefore, this paper did not analyze the causes and mechanisms of the spatial pattern of the factors such as economy, age, gender, land price, education, and transportation. In addition, this paper focused on the quantitative analysis and lacked a qualitative evaluation of the sports facilities’ layout. Finally, when we constructed the social performance model, the factors such as age, gender, and income were not considered. Future research can continue to optimize the model.

## 5. Conclusions

Through spatial analysis and social performance evaluation, this paper studied the temporal and spatial evolution pattern of Shanghai’s community sports and fitness venues from 1982 to 2019. The results may help improve the fairness of urban public services and the quality of residents’ lives and promote the implementation of the “Healthy China” national strategy. The main conclusions are as follows: (1) The spatial and temporal evolution of community sports and fitness venues in Shanghai follows the pattern of “urbanization-suburbanization-reurbanization”. The construction focus has always been in the urban area but has moved toward “first south and then north”. Government investment has always been the main source of funding for the construction of community sports and fitness venues in Shanghai. Social capital investment has also steadily grown. Citizens’ educational fitness centers have always been the main type of community sports and fitness venues, but the number of multiple types of venues, such as trails, gymnasiums, and courts, has increased significantly. There has been an increase in the construction area. Attention has been given to diversified and personalized sports and fitness needs. (2) The overall service radius coverage rate of Shanghai’s community sports and fitness venues and the regional equality between the core and peripheral areas have been significantly improved. The overall per capita service location entropy has not improved significantly. The old city center and the peripheral area have always been low-value areas. The old city center is surrounded by high-value areas. The “low-high-low” three-circle spatial structure continues to exist, but the high-value area around the old city center expanded significantly and achieved greater social performance optimization from 1999 to 2009. (3) The social performance of Shanghai’s community sports and fitness venues in the urban area is better than that in the suburban areas, but the optimization of social performance in the suburban areas is greater than that in the urban areas.

Future research on the layout of urban sports and fitness venues can be strengthened in the following ways: (1) Exploration of the causes and mechanisms of the social performance’s evolution to put forward more specific suggestions. (2) Construction of a more comprehensive social performance evaluation system by considering the demands of different groups. (3) Comparison of social performance in cities with different economic, cultural, and political backgrounds to promote the development of urban planning. (4) Study of urban sports facilities from a micro scale by considering their operation and facility resources, such as available services, target population, and other details.

According to the research conclusions, we propose the following policy suggestions: First, the layout of community sports and fitness venues needs to be improved, and the number of those in suburban areas needs be increased to a proper level. Second, the government should control the excessive growth of the urban population and improve the social equity of community sports and fitness services in the old city center. Third, we should encourage social investment in construction and operation. By adopting these strategies, we may enrich the types of sports and fitness venues to meet the residents’ diverse and personalized needs.

## Figures and Tables

**Figure 1 ijerph-19-00274-f001:**
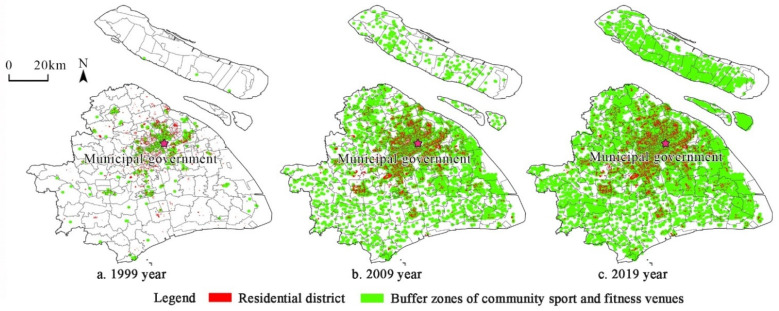
Service range of Shanghai’s community sports and fitness venues in 1999, 2009, and 2019.

**Figure 2 ijerph-19-00274-f002:**
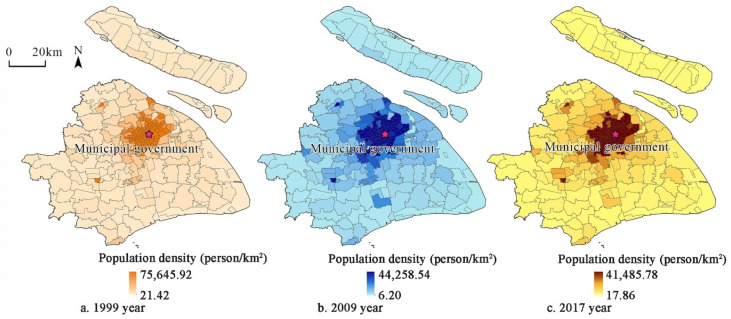
Shanghai’s population distribution in 1999, 2009, and 2017.

**Figure 3 ijerph-19-00274-f003:**
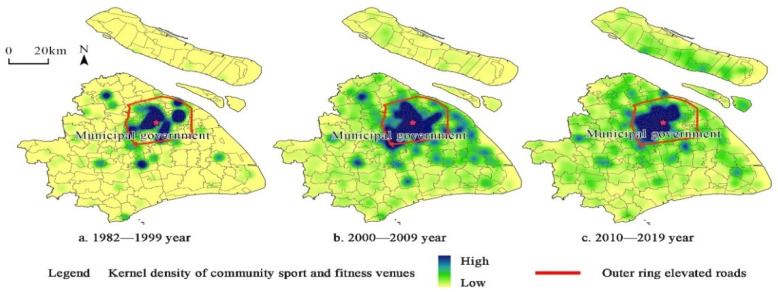
Spatial-temporal evolution of kernel density of Shanghai’s community sports and fitness venues.

**Figure 4 ijerph-19-00274-f004:**
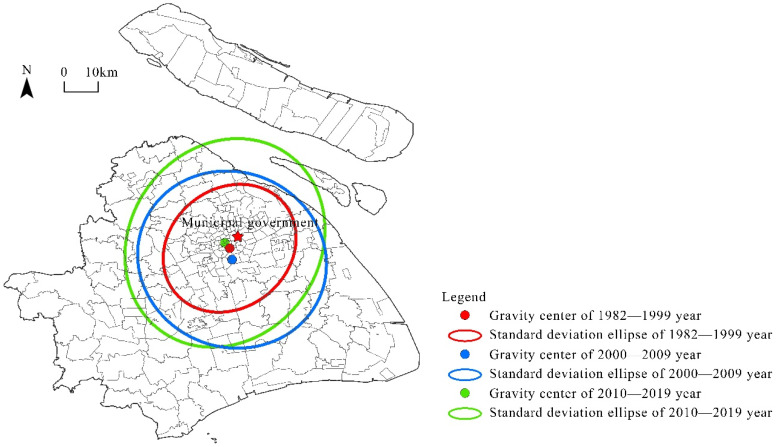
Evolution of the standard deviational ellipse distribution of Shanghai’s community sports and fitness venues.

**Figure 5 ijerph-19-00274-f005:**
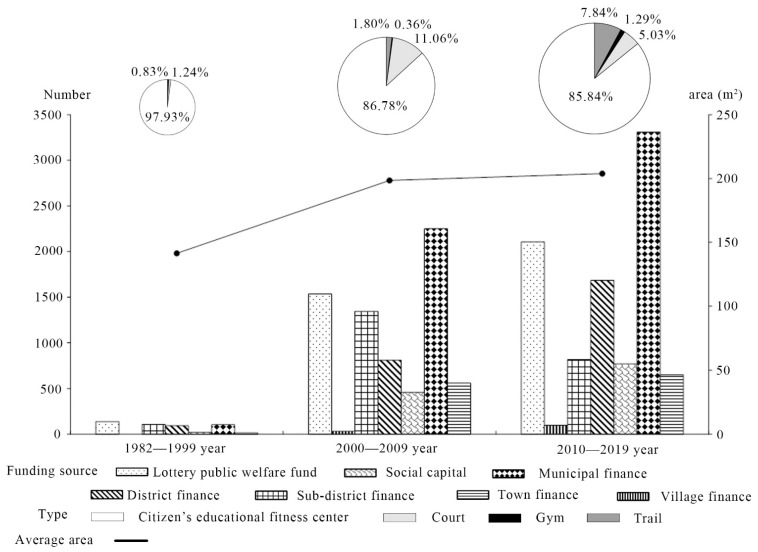
Evolution of the attributes of Shanghai’s community sports and fitness venues.

**Figure 6 ijerph-19-00274-f006:**
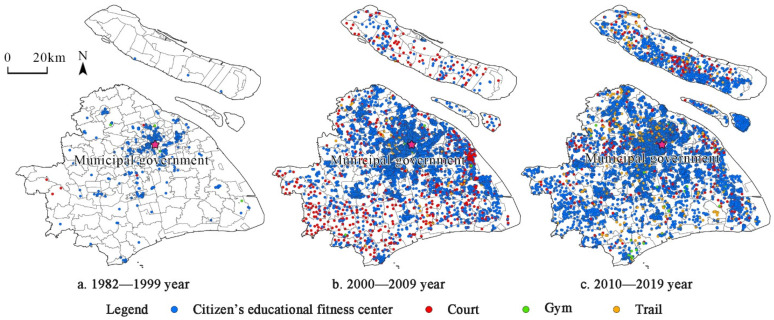
Evolution of the types of Shanghai’s community sports and fitness venues.

**Figure 7 ijerph-19-00274-f007:**
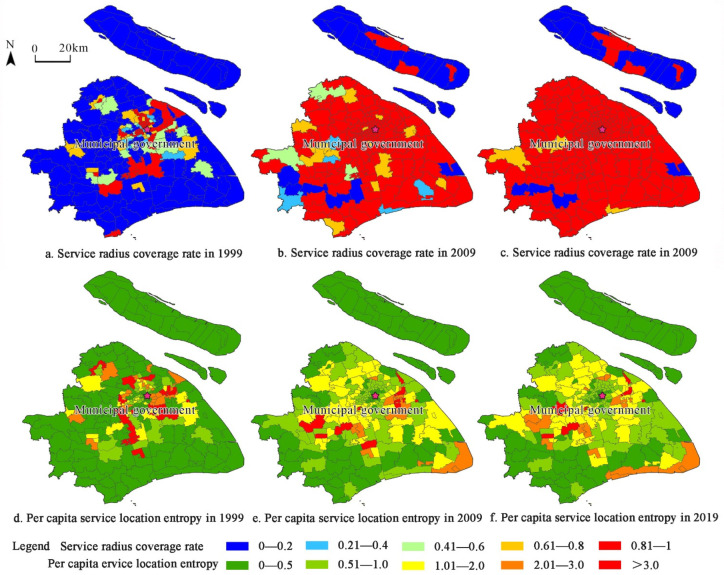
Evaluation of the social performance of Shanghai’s community sports and fitness venues.

**Table 1 ijerph-19-00274-t001:** Differences in social performance of community sports and fitness venues between urban areas and suburban areas.

Items		Urban Area	Suburban Area	Whole Area
Service radius coverage rate	mean of 1999	0.53	0.17	0.33
	mean of 2009	0.98	0.72	0.84
	mean of 2019	1	0.8	0.89
Per capita service location entropy	mean of 1999	1.4	0.86	1.11
	mean of 2009	1.28	0.91	1.08
	mean of 2019	0.92	0.89	0.91

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
