# Peer review of "Evolution of the Spatial-Temporal Pattern and Social Performance Evaluation of Community Sports and Fitness Venues in Shanghai"

_ijerph, 2021, doi:10.3390/ijerph19010274_

Round 1
Reviewer 1 Report
It is very hard to refere the situation in China to Western countries, without showing more the political and economic conditions, which differs.
In most of European countries only small amount of sport facilities are financed by the state/city, therefore the input of private investors is very important.
Discussion should be widely enlarged.
Basically, there were no conclusions presented.
Reviewer 2 Report
Thank you for the opportunity to review this interesting paper.
This paper analyzes changes in the distribution of community sport and fitness venue in Shanghai and discusses them in terms of regional equality and social equity. Regional equality and social equity are analyzed by service radius coverage rate and per capita service location entropy, respectively, and these methods are appropriate. The analysis results of spatiotemporal patterns provide certain findings for the evaluation of urban service facilities.
If the following revise is made, it will be easier for the reader to understand the significance of this paper.
line 158
Figure 1 should improve the resolution. The relationship between the residential district and buffer zones is difficult to understand.
line 176-178
The authors should clearly describe the academic or policy value of studying temporal and spatial evolution.
line 190-191
What is the unit of data for residential district? Please provide the definition of residential district in this paper.
line 262-263
The period is divided into three phases: 1982-1999, 2000-2009, and 2010-2019, but the data in 1982 is not used. Why is the start of the period set in 1982? In addition, the description that “there were few community sport and fitness venues in the 1980s” requires a reference. Figures 3 and 5 do not show the changes, so these figures should be represented as the results in the three years.
line274-280
Regarding the construction trends from 2000 to 2009, it seems that the description "mainly in urban area" and the description "significantly in suburban areas" are inconsistent.
line 288-297
What is the necessity of investigating the shift of gravity center of overall spatial distribution
using the standard deviational ellipse method?
It is understandable that the gravity center represents a deviation in the spatial distribution of the facility, but its significance for equality and fairness is not clear.
Besides, the place name of the concrete center does not seem to have any meaning.
The name of the center districts does not seem to have any meaning.
line 422-424
In this paper, a quantitative evaluation of sports facilities was conducted, but a qualitative evaluation such as the richness or layout was not evaluated. Therefore, the statement " has made considerable progress in terms of the richness of types, and the optimization of the layout " is not appropriate.
Reviewer 3 Report
The manuscript, in which the authors examined evolution spatio-temporal pattern and social performance evaluation of community sports and fitness venues in Shanghai, is a local case study and its novelty is low. The manuscript does not make an innovative contribution to the literature. Furthermore, the main purpose and the importance of the study were not defined and stated clearly. The Conclusions are weak and insufficient, thus the section must be improved. Finally, the Discussion section should be located top of the Conclusion. All in all, the manuscript cannot be published in the journal IJERPH as such.
Reviewer 4 Report
Interesting article: however, it would be helpful to present a historical discussion of sports venues in the city and how it has transformed to where they are today. This would help to see if the changes reported in the article are new or a part of the long-term changes.
Defining what each of the sports venue are in terms of available services, target population and other details can contextualize the location of venues in the city. I believe the venues vary in their operation and facility resources.
This research is only exploring the distribution of the venues in the city without any relationship to factors such as policies, population migration pattern, economic changes, ageing, education, and other variables that would affect how residents use them and likely related to where new facilities are built in the city.
The maps are not clear in terms of what they are displaying. Label them appropriately or add more legends to distinguish differences in the data. For example, population density should be listed per area or create a dot map of the location of venues by type in the city.
Round 2
Reviewer 1 Report
The authors have largely changed the initial manuscript, introducing new paragraphs, according to the initial demand from the 1st stage of review.
The reviewer apprieciate the amount of work done by the authors and reccomend to publish the manuscript in the present form.
Reviewer 3 Report
I would like to thank the authors for considering all of my suggestions. I have no objections to the corrections made, but the importance and significance of the paper should be emphasized in the Introduction instead of the Discussion section.
